# Association of Maternal Antenatal Education with Quality of Life after Childbirth in the Slovenian Population before and during the COVID-19 Pandemic

**DOI:** 10.3390/healthcare11111568

**Published:** 2023-05-26

**Authors:** Tina Berčan, Nina Kovačević, Ines Cilenšek, Iztok Podbregar

**Affiliations:** 1Emergency Medical Dispatch Service, Emergency Medical Dispatch Centre Ljubljana, University Medical Centre Ljubljana, 1000 Ljubljana, Slovenia; 2Faculty of Medicine, University of Ljubljana, 1000 Ljubljana, Slovenia; nkovacevic@onko-i.si; 3Institute of Oncology Ljubljana, 1000 Ljubljana, Slovenia; 4Institute of Histology and Embryology, Faculty of Medicine, University of Ljubljana, 1000 Ljubljana, Slovenia; 5Faculty of Organizational Sciences, University of Maribor, 4000 Kranj, Slovenia

**Keywords:** pregnancy, antenatal classes, quality of life

## Abstract

Pregnancy and childbirth have a crucial impact on a woman’s quality of life. In Slovenia, antenatal classes are the main educational tool used to prepare expectant mothers for their new role. The aim of our study was to assess the relationship between the duration of antenatal classes and the mothers’ quality of life after childbirth. A self-administered, previously validated and tested questionnaire regarding the quality of life after childbirth was completed by Slovenian women. Based on an online survey, data were collected for two groups of mothers. The first group (*n* = 1091) gave birth before the COVID-19 pandemic, and the second group (*n* = 1163) gave birth during the pandemic. Group differences were analyzed using the Mann–Whitney U test. Linear regression and correlation coefficients were calculated for the association between quality of life and the duration of antenatal classes. Our study showed a significant decrease in the duration of antenatal classes and a decrease in quality of life after birth during the COVID-19 pandemic. We also showed that more antenatal education was associated with a higher quality of life. Despite the influence of multiple factors during the COVID-19 pandemic, we defined the correlation between the duration of antenatal classes and postpartum quality of life in a sample of Slovenian mothers. The duration of the antenatal classes is an important factor influencing the quality of life after childbirth.

## 1. Introduction

The postpartum period and quality of life after birth are important factors in women’s lives. The peculiarity of the postpartum period is that at the time of the child’s birth, the mother and father assume responsibility for new family members who are unable to care for themselves. All this leads to numerous emotional and social changes [1]. In couples without antenatal education, the risk for personality and partnership problems might be higher, which can be reflected in the child’s attachment and emotional development [2]. After childbirth, women adapt to the maternal role and experience changes in the physical, psychological, social, and familial domains [3]. These changes, the perception of the changes, and the fulfilment of expectations influence the assessment of postpartum quality of life [4]. Setting realistic expectations and reducing discrepancies between expectations and the subsequent reality are fundamental to the postpartum quality of life. The reality of expectations can be influenced by adequate education about the possible courses of childbirth and ongoing changes in the postpartum period [3].

In Slovenia, antenatal classes represent the main method of educating and preparing mothers/couples for childbirth and the postpartum period. Antenatal classes in Slovenia are led by health care professionals, such as nurses or midwives, and obstetricians. They are designed to help expectant parents prepare for birth, infant care, and parenthood. These classes typically cover a wide range of topics, including the stages of birth, pain management options, infant feeding, diapering, bathing, and safety. Attending antenatal classes during pregnancy is not mandatory. The decision to attend antenatal classes is up to the women themselves, and they also currently have access to various online platforms and mobile applications that can be used independently [5].

Our previous study in a population of 1043 pregnant women showed that 79% of women who gave birth between March 2018 and March 2019, and 70% of their partners, attended antenatal classes [6]. Hong demonstrated the importance of prenatal education, as mothers who received information about childbirth and postpartum changes reported a significantly higher postpartum quality of life [7]. Rezaei demonstrated the positive influence of education and support from all types of medical personnel. He also showed that maternal quality of life was significantly influenced by health care visits and confirmed that role of health care professionals in providing education and support related to child care and postpartum changes [8]. Cutajar showed that adequate prenatal education by health care professionals improves the acceptance of pregnancy and its associated changes and reduces fear of childbirth, but does not change perceptions of motherhood [9]. Delicate showed that the transition to parenthood is a stressful event for both partners, which has a negative impact on quality of life. He emphasized the importance of prenatal medical care and the significance of antenatal classes in preparing the couple for the changes occurring after the birth of the child. The positive influence of education regarding the physical and psychological changes that affect quality of life has been proven [1]. Gun Kakasci’s research showed that majority first-time mothers were better prepared for childbirth and early parenthood as a result of antenatal education provided by medical personnel [2]. Lundquist and Marshall showed that medical personnel working in the outpatient department and nursing were instrumental in providing emotional support to pregnant women and first-time mothers through education and explanations [10,11]. Swift and Munkhondya found that prenatal education reduced the fear of childbirth in pregnant women and increased the mother’s self-reliance in preparing for birth and transitioning to motherhood [12,13].

The COVID-19 pandemic has led to many changes in field of work, employment, education, social support, loneliness, marriage status, social disruption, bereavement, work environment, social status, social integration, and people’s psychosocial perceptions of life priorities and overall quality of life [14]. Tsuno showed that women who gave birth and raised a child during the COVID-19 pandemic had a higher incidence of of postpartum depression, which they associated with less social support from medical personnel and the wider family circle [15]. Aksoy showed the positive impact of distance education on reducing stress and anxiety, both during pregnancy and childbirth [16]. Preis showed that due to the epidemiological restrictions during the COVID-19 pandemic, satisfaction with the birth experience was lower among pregnant women, which was mainly due to the absence of a partner during childbirth, as well as less access to information regarding the process and decreased readiness of women for the actual act of childbirth [17]. Gonzalez-Garcia showed that during the COVID-19 pandemic, there was greater concern among pregnant women and women with children about the child’s well-being, social support, and postpartum care [18]. Guner showed that the COVID-19 pandemic and the associated restrictive measures had many effects on the lives and well-being of pregnant women, including effects on social life and physical well-being, as well as effects on psychological health, including anxiety, fear of childbirth, and concerns regarding newborn health care [19].

From the results of the above studies, we can conclude that the COVID-19 pandemic had a multidimensional impact on people’s lives. It also affected the Slovenian antenatal classes as an established method of education, as the total duration of the antenatal classes was shortened. The aim of the study was to investigate the relationship between the duration of antenatal classes and postpartum quality of life in Slovenian women. In addition, the study aimed to evaluate the importance of the duration of organized and professional knowledge transfer during these classes.

## 2. Materials and Methods

### 2.1. Study Design

A cross-sectional study, using a previously validated and translated questionnaire, was conducted in Slovenia between May 2018 and May 2019 (before the COVID-19 pandemic) and between May 2020 and May 2021 (during the COVID-19 pandemic, to evaluate the quality of life after childbirth.

### 2.2. Study Population and Sample Size

Slovenian women who attend antenatal classes in Slovenia were eligible to participate in the study. We included 2254 unrelated Slovenian (Caucasian) pregnant women. In the control group (G1), we enrolled 1091 women who gave birth between March 2018 and March 2019 (before the COVID-19 pandemic). In the study group (G2), we enrolled 1163 women who gave birth between May 2020 and May 2021 (during the COVID-19 pandemic).

The sample collected was considered representative, as it denotes about 5.5% of all annual births in Slovenia (Table 1) [20].

### 2.3. Participant Recruitment

The survey was posted online on social media platforms, including Facebook and Instagram, as well as online Slovenian forums for expectant mothers in order to reach a wider audience. Mothers were interviewed using a self-administered questionnaire compiled using the 1KA tool (https://www.1ka.si/, accessed on 1 March 2018). The questionnaire was pretested on 15 respondents in the form of a pilot study. The reliability of the measurement tool, which was related to postpartum quality of life, was checked using the Cronbach (α)-index, which was 0.728, indicating that the questionnaire showed good reliability. A convenience sampling technique was used to select women who met the study’s inclusion criteria and completed the questionnaire. The inclusion criterion required at least a 90% attendance (measured in minutes) at antenatal classes in Slovenia during pregnancy. An exclusion criterion included an incomplete response to the questionnaire. Women were informed about the purpose and objectives of the study. Participation in the study was voluntary and anonymous. All data were treated confidentially.

### 2.4. Research Tools

A self-administered, previously validated, translated (into the Slovenian language) and tested regarding the quality of life after childbirth questionnaire (perceived quality of life scale—PQOL) (Appendix A Appendix A) [21] was completed by Slovenian women. The PQOL is a specific questionnaire designed for postpartum women. In addition to completing the PQOL questionnaire, mothers were also asked questions to obtain sociodemographic and obstetric data. A comparison of the quality of life results was performed for the period before and during the COVID-19 pandemic.

To assess postpartum quality of life, we used an online questionnaire that had already been validated in the Slovenian population [6]. The PQOL questionnaire assesses individual segments of quality of life, namely child care, physical well-being, psychological well-being, and social involvement and support of mothers. The questionnaire includes 40 questions divided into four thematic groups: Child Care, Physical Well-Being, Psychological Well-Being, and Social Inclusion and Support. Child Care includes 8 questions, Physical Well-Being includes 12 questions, Psychological Well-Being includes 8 questions, and Social Inclusion and Support includes 12 questions. The respondents’ opinions and attitudes were assessed using a five-point Likert scale (1–5) with which respondents indicated the extent to which they agreed with the question asked. The higher the number, the greater the agreement with the statement, or the better the mother’s postpartum condition.

A structured interview was prepared for the organizers of antenatal classes in all 14 Slovenian regional hospitals. The interview aimed to gather information regarding the type (in person or virtual) and duration (in minutes) of lectures conducted before and during the COVID-19 pandemic. The interviews were conducted by two trained researchers, I.P. and T.B., who had prior experience in qualitative research and were familiar with the study topic. We also checked to determine whether all antenatal classes followed a uniform topic plan that summarized all necessary information, including information about pregnancy, birth, changes in the postpartum period, and care of the newborn.

### 2.5. Statistical Analysis

For statistical analysis, data were analyzed using SPSS, version 26 (IBM Inc., Armonk, NY, USA). Continuous sociodemographic and obstetric data were compared with the unpaired Student’s *t*-test, whereas the Chi-squared test (χ^2^) was used to compare categorical data. Data were presented as mean ± SD (continuous variables) or as the number and percentage of women (categorical variables). To assess the statistical significance of the obtained PQOL results, the answers for each quality of life segment (Child Care, Physical Well-Being, Psychological Well-Being, Social Involvement and Support) were summed and compared using the Mann–Whitney U test before and during the COVID-19 pandemic. The Mann–Whitney U test was also used to estimate the duration of antenatal classes during the two periods.

The collected data regarding the quality of life of each segment covered by the PQOL (Child Care, Physical Well-Being, Psychological Well-Being, Social Involvement and Support) were divided into 5 categories, based on the maximum possible score. For the Child Care segment (points range: 8–40 points), the categories were as follows; category 1: 8–13 points, category 2: 14–20 points, category 3: 21–27 points, category 4: 28–34 points, and category 5: 35–40 points. For the Physical Well-Being segment (points range: 12–60 points), the categories were as follows; category 1: 12–21 points, category 2: 22–30 points, category 3: 31–40 points, category 4: 41–50 points, and category 5: 51–60 points. For the Psychological Well-Being segment (points range: 8–40 points), the categories were as follows; category 1: 8–13 points, category 2: 14–20 points, category 3: 21–27 points, category 4: 28–34 points, and category 5: 35–40 points. For the Social Involvement and Support segment (points range: 12–60 points), the categories were as follows; category 1: 12–21 points, category 2: 22–30 points, category 3: 31–40 points, category 4: 41–50 points, and category 5: 51–60 points.

In the statistical analysis of the postpartum quality of life (PQOL) segments, it was originally intended to designate category No. 3 (“the group in the middle”) as the reference category. However, due to the small number of participants in certain categories, statistical analysis could not be conducted separately for each category. Therefore, to ensure the inclusion of all participants and enable statistical analysis, those categories with small numbers were merged with the closest group in terms of points. As a result of this merging process, the number of the reference categories changed accordingly, but remained the same from a points perspective. This adjustment was made to ensure a robust statistical analysis that considers the complete participant sample.

The higher the categorical number, the higher the quality of life, based on the PQOL scoring. Linear regression analysis was used to quantify the relationship between the duration of antenatal classes and all four segments of quality of life. We performed a multinomial logistic regression analysis to assess the association between all four segments of quality of life and the duration of antenatal classes, as well as sociodemographic and obstetric data. A multiple correlation model with Spearman’s coefficient was used to obtain a correlation estimation between quality of life and other collected data regarding demographics and childbirth. A value of *p* < 0.05 was considered statistically significant.

### 2.6. Ethical Approval

The study was approved by the Institutional Review Board, Faculty of Organizational Sciences, and the Senate of the University of Maribor (Approval No. 012/2018/1).

## 3. Results

The questionnaire was fully completed by 1091 women who gave birth between March 2018 and March 2019—group 1 (G1), and by 1163 women who gave birth between May 2020 and May 2021—group 2 (G2). Both groups were matched by age, education level, marital status, economic status, monthly household income, place of residence, consecutive birth, type of birth, and postpartum grief/anxiety, as there was no statistically significant difference between G1 and G2 in regards to these parameters (Table 2).

The duration of antenatal classes was statistically significantly shorter in G2 compared to G1 (*p* < 0.001). The average duration of antenatal classes was 399 min in G1 and 244 min in G2. Before the COVID-19 pandemic, antenatal classes were held in person (G1); however, during COVID-19, all antenatal classes were held online (G2).

The comparison was made for each of the quality of life segments covered in the PQOL questionnaire (Child Care, Physical Well-Being, Psychological Well-Being, and Social Inclusion and Support). The median scores of the PQOL segments were compared, and a statistically significant difference in quality of life between G1 and G2 was found, in favor of G1 in all four quality of life segments of the PQOL (Table 3).

The linear regression model showed statistical significance for the test models of all four segments of quality of life. Further analysis showed that antenatal classes impact quality of life. For the Child Care quality of life segment, the B coefficient showed that for each additional minute of antenatal classes, the quality of life in this segment increased by 0.028 points. For the Physical Well-Being quality of life segment, the B coefficient showed that for each additional minute of antenatal classes, the quality of life in this segment increased by 0.038 points. For the Psychological Well-Being quality of life segment, the B coefficient showed that for each additional minute of antenatal classes, the quality of life in this segment increased by 0.037 points. For the Social Inclusion and Support quality of life segment, the B coefficient showed that for each additional minute of antenatal classes, the quality of life in this segment increased by 0.051 points (Table 4).

The aim of this study was to examine the relationship between quality of life, sociodemographic and obstetric data, and the duration of antenatal classes using multinomial logistic regression. Analysis of categorically classified segments of quality of life showed the association with other categorical data collected (Table 5, Table 6, Table 7 and Table 8). The model fitting information in the statistical analysis showed that the model containing a full set of predictors showed a significant improvement over the null. The likelihood ratio tests showed a significant effect of the duration of antenatal classes, economic status, monthly household income, and consecutive birth in the model used, so the total contribution of each of the above independent variables to the model was statistically significant. For the purpose of multinomial logistic regression analysis, PQOL outcomes were divided into 5 categories for each quality of life segments.

For the segment Child Care quality of life, there were only 5 women in category 1, so categories 1 and 2 were combined into category 1. The reference category was category number 2. Statistical significance was found in the categories of antenatal classes duration and place of residence. The odds ratio showed that a higher level of education and a rural residence (population less than 3000) decreased the probability of inclusion in category 1 and increased the probability of inclusion in categories 3 and 4 (higher quality of life). Duration of antenatal classes was a significant predictor (B −0.004; *p* < 0.001) in the model and showed that the longer the duration of antenatal classes, the higher the probability of being classified in category 4 (higher quality of life). Other data collected showed no statistically significant effect on the Child Care segment of quality of life.

For the segment Physical Well-Being quality of life, in category 5, there was only one female, so categories 4 and 5 were combined into category 4. The reference category was category 3. Statistical significance was found in the categories of the duration of antenatal classes, economic status, and consecutive births.

The odds ratio showed that a higher educational level decreased the probability of inclusion in categories 1 and 2 and increased the probability of inclusion in category 4 (higher quality of life). The odds ratio for economic status showed that employed women were less likely to be included in categories 1 and 2. The higher the number of consecutive deliveries, the higher the odds for women to be classified in category 1 and 2. Rural residence (population less than 3000) was also a predictor of classification in category 4 (higher quality of life). Attendance at antenatal classes was a significant predictor (B −0.026; *p* < 0.001) in the model, showing that the longer the duration of antenatal classes, the higher the odds of being classified in category 4 (higher quality of life). Other data collected showed no statistically significant impact on Physical Well-Being quality of life.

For the segment Psychological Well- Being quality of life, there were only 9 women in category 2, so categories 1 and 2 were combined into category 1, and there were only 4 women in category 5, so categories 4 and 5 were combined into category 3. The reference category was category number 2. Statistical significance was found in the categories of the duration of antenatal classes, economic status, and consecutive births.

The odds ratio showed that a higher level of education decreased the women’s odds of being included in category 1 and increased their odds of being included in category 3 (higher quality of life). Antenatal classes were a significant predictor (B −0.002; *p*= 0.003) in the model and showed that the longer the antenatal education time, the higher the odds of being included in category 3 (higher quality of life). The odds ratio for economic status showed lower odds of inclusion in category 1 for employed women and higher odds of inclusion in category 3 (higher quality of life). The higher the number of consecutive deliveries, the higher the odds for women to be classified in category 1 (lower quality of life). Other data collected showed no statistically significant effect on Psychological Well-Being quality of life.

For the segment Social Inclusion and Support quality of life, the reference category was category number 3. Statistical significance was found in the categories of the duration of antenatal classes, economic status, monthly household income, number of consecutive deliveries, and the presence of the father at the birth.

The odds ratio showed that a higher educational level decreased the probability of inclusion in category 1 and 2 and increased the probability of inclusion in category 4 (higher quality of life). Antenatal classes were a significant predictor (B −0.027; *p* < 0.001) in the model and showed that the longer the antenatal education time, the higher the odds of being classified in category 5 (higher quality of life). The odds ratio for economic status indicated a lower probability of being classified in category 1 for employed women. The father’s presence at the birth decreased the odds of classification in category 1. As monthly household income increased, the probability of classification in category 2 was lower and the probability of classification in category 4 was higher, implying that higher household income correlated with better social inclusion and support. A higher number of consecutive deliveries also indicated a higher probability of classification in categories 1 and 2 (lower quality of life).

The correlation between each segment of quality of life and the duration of antenatal classes was assessed (Table 9). Because of the non-normal distribution, the Spearman’s correlation test was used. It showed a statistically significant correlation between all segments of quality of life. Based on the grading standards, the correlation between the Child Care segment and the duration of antenatal classes was moderate, the correlation between the Physical Well-Being segment and the duration of antenatal classes was weak, the correlation between the Psychological Well-Being segment and the duration of antenatal classes was weak, and the correlation between the Social Inclusion and Support segment and the duration of antenatal classes was moderate.

## 4. Discussion

It has previously been shown that antenatal education is an important element that influences postpartum events for the mother [22,23,24,25,26,27]. Based on the literature review, our study is the first to evaluate the association between the duration of antenatal classes and women’s postpartum quality of life in such a large population sample.

Our study showed that the duration of antenatal classes during the COVID-19 pandemic was significantly shorter compared to the length of classes before the COVID-19 pandemic in Slovenia (399 vs. 244 min, *p* < 0.001). With a shorter duration of antenatal classes, the quality of life during the COVID-19 pandemic decreased significantly. Other authors did not report on the duration and possible changes in the duration of antenatal classes. An Iranian study showed that antenatal classes contributed to a higher quality of life regarding physical well-being, psychological well-being, and environmental health among first-time pregnant women [28].

Before the COVID-19 pandemic in Slovenia, antenatal classes were held in person, but during COVID-19, due to health care restrictions, antenatal classes were moved online. This change in the presentation of antenatal classes opened the possibility for biases, despite the uniformity of all Slovenian antenatal education programs. An online form of antenatal classes can be a good substitute for an in-person class for pregnant women who need basic information about pregnancy, childbirth, newborn care, and the postpartum period. Wu showed that the virtual form of education was a popular form of education among pregnant women during the COVID-19 pandemic because they received basic useful information about pregnancy and childbirth. In addition, they were able to consult with medical staff remotely. It should also be emphasized that the virtual form of education is more economically efficient and has also enabled socially weaker pregnant women to access information. Some pregnant women doubted the reliability of such online information [29]. Chen showed that during the COVID-19 pandemic, virtual forms of antenatal classes were well accepted and frequently used by pregnant women. The use of virtual classes was higher during the COVID-19 pandemic than before the pandemic. The online antenatal class contributed significantly to reducing nonemergency hospital visits and pregnant women’s exposure to possible infection with COVID-19 [30]. Uludag showed that online antenatal classes offered during the COVID-19 pandemic reduced worries about birth, fear of childbirth, and anxiety about COVID-19 and improved readiness for birth [31].

Our study showed that there was a statistically significant difference in quality of life between a group of mothers who gave birth before the COVID-19 pandemic and a group of mothers who gave birth during the COVID-19 pandemic in all four segments regarding quality of life. The quality of life of Slovenian women was lower during the COVID-19 pandemic.

Our study showed that higher education and rural residence (with a population less than 3000) were associated with higher quality of life in the Child Care segment. Japanese and Iranian studies have also shown the importance of prenatal education in the Child Care segment for decreasing maternal postpartum depression [27,32]. Tsuno showed that women who had fewer opportunities to consult health professionals also attended fewer newborn checkups, had less support from the fathers and other relatives, and therefore had a lower quality of life and were more likely to experience postpartum depression [15].

The Physical Well-Being and Psychological Well-Being segments of quality of life are better in women with higher education and a stable job. However, an increase in the number of consecutive deliveries decreased the quality of life in those two segments. A French study examined anxiety, posttraumatic stress symptoms, and emotion regulation in pregnant women who gave birth during the COVID-19 pandemic. The level of worry about the baby was found to be negatively associated with the level of social support. The study also showed that the greater the concern for the baby, the higher the anxiety and depression scores [18]. An American study showed that maternal self-efficacy and social support were negatively associated with postpartum parenting stress. COVID-19 restrictions and maternal depression and anxiety also negatively affected maternal self-efficacy. The study also showed that in cases where maternal self-efficacy was high, other negatively influencing parameters were attenuated [33].

We also found that more education, a stable job, the father’s presence at the birth, and increased household income were associated with a higher quality of life in the Social Inclusion and Support segment. The benefits of antenatal classes include increased knowledge and confidence for expectant parents, a better understanding of the birth process, and better preparation for the physical and emotional demands of childbirth and early parenthood. In addition, attending antenatal classes can provide expectant parents with the opportunity to share information with others who have had similar experiences, fostering a sense of community and support [34,35,36]. A Swedish study showed that antenatal classes whose content focused on birth and parenthood helped parents to better manage various birth and post-birth situations [26].

Our research has shown a positive correlation between prenatal education and quality of life for expectant mothers. This is consistent with previously published studies that have shown that that antenatal education is associated with a better quality of life after birth [23,37]. Based on grading standards, we found that the correlation between the Child Care and Social Inclusion and Support segments of quality of life and the duration of antenatal classes was moderate, and the correlation between the Physical Well-Being and Psychological Well-Being segments of quality of life and the duration of antenatal classes was weak. We found that for each additional minute of antenatal classes, each of the quality of life segments would increase by a point factor. A Polish study observed the level of depression and anxiety in pregnant women and found that women who attended antenatal classes in person had the lowest levels of anxiety and depression, followed by those attending online classes, and finally, those attending no classes at all [37]. Chen analyzed the use of an online mobile app in prenatal care and found that the online app was a well-used educational tool during the COVID-19 pandemic [30]

The quality of life in all studied segments of PQOL seems to be a multifactorial issue. Our study contributes to the evaluation of the importance of the duration of antenatal classes by trying to quantitatively assess its correlation with quality of life and also evaluate the influence of other possible factors on the quality of life of postpartum women in Slovenia. The study contributed to the understanding of the different segments of quality of life and the possible opportunities for educational health professionals to play an active role in improving the quality of life of postpartum women. Our study shows that a decrease in information and the extent of education provided during pregnancy is directly related to a lower quality of life after childbirth. It seems legitimate to assume that planning interventions to promote women’s health at this sensitive period of life will have positive long-term effects. This is consistent with Wu’s conclusions that it is important to ensure the quality and safety of online services and to establish a stable, mutual trust between pregnant women, obstetric care providers, and online education programs [29].

Given the limited or nonexistent influence of health personnel on sociodemographic parameters, it is imperative that at least the educational measures fulfil their role to the maximum extent possible, meaning ensuring a sufficient duration of antenatal classes.

Various limitations apply to our research. Our study was conducted on the Slovenian population. When conducting a study in a particular country or cultural context, it is imperative to acknowledge that the results may be influenced by unique cultural, social, economic, or historical factors specific to that particular context. Therefore, caution must be exercised when attempting to extrapolate results to other countries or cultures, as these factors may differ significantly and may potentially affect the relationships between variables. It is important to recognize that further research in other countries or cultural contexts is necessary to determine the generalizability of the results. Although the results of our study are statistically significant, the actual impact of these variables may be limited in a real-world context. Statistically significant results indicate that there is likely to be a relationship between variables, but this does not necessarily mean that the effect size is substantial or practically important. In the case of studies of small magnitude, it is important to recognize this. The statistical significance of the results may be due to the large sample size used in this study, which increases the probability of detecting even small effects. This should be acknowledged as a limitation because the practical implications of such a small effect size might be of limited importance. Prior to the COVID 19 pandemic antenatal classes were held in person; during the COVID 19 pandemic, they were held online due to health restrictions. Online classes might lack engagement with the participants, and some women might be difficult to accommodate online. This may also affect the results.

## 5. Conclusions

We anticipate that our research will make an important contribution to elucidating problems and changes in the postpartum period, suggesting specific possibilities for an active role of antenatal classes in improving the quality of life after birth. Planning interventions to provide adequate education, improve quality of life, and promote women’s health during this sensitive period of life brings favorable long-term effects. Considering the limitations we observed, our study nevertheless lays the groundwork for further evaluation of the duration of antenatal classes in relation to quality of life and points to potential organizational limitations.

## Figures and Tables

**Table 1 healthcare-11-01568-t001:** Number of births per year in Slovenia.

Year	Number of Births
2018	19,123
2019	18,946
2020	18,432
2021	18,692

**Table 2 healthcare-11-01568-t002:** Sociodemographic and obstetric data.

Sociodemographic and Obstetric Variables	Group 1 *n* = 1091	Group 2 *n* = 1163	*p*-Value
Age (years)	29.21 ± 4.28	29.45 ± 4.07	0.186
Level of education			
*Elementary school*	13 (44.8%)	16 (55.2%)	0.858
*High school*	351 (48.5%)	373 (51.5%)	
*College*	251 (46.7%)	287 (53.3%)	
*University*	411 (49.6%)	417 (50.4%)	
*Master of Science, Ph.D*	65 (48.1%)	70 (51.9%)	
Marital status			
*Married*	440 (48%)	476 (52%)	0.373
*Divorced*	6 (46.2%)	7 (53.8%)	
*Non-marital partnerships*	632 (48.4%)	674 (51.6%)	
*Single*	13 (68.4%)	6 (31.6%)	
Economic status			
*Employed*	689 (48.2%)	741 (51.8%)	0.843
*Self-employed*	57 (46.3%)	66 (53.7%)	
*Unemployed*	270 (49.2%)	280 (50.8%)	
*Single*	75 (49.7%)	76 (50.3%)	
Monthly household income			
*Less than EUR 700*	43 (50.6%)	42 (49.4%)	0.572
*EUR 701–1500*	401 (50.1%)	399 (49.9%)	
*EUR 1501–2500*	426 (48.2%)	458 (51.8%)	
*EUR 2501–4000*	191 (45.2%)	232 (54.8%)	
*More than EUR 4001*	30 (48.4%)	32 (51.6%)	
Place of residence			
*More than 3000 inhabitants*	546 (50.1%)	544 (49.95)	0.121
*Less than 3000 inhabitants*	545 (46.8%)	619 (53.2%)	
Consecutive birth			
*First*	661 (49.4%)	677 (50.6%)	0.548
*Second*	336 (48.1%)	363 (51.9%)	
*Third*	79 (43.6%)	102 (56.4%)	
*Fourth*	12 (40%)	18 (60%)	
*Fifth or greater*	3 (50.0%)	3 (50.0%)	
Mode of birth			
*Vaginal* *birth*	853 (49.2%)	879 (50.8%)	0.341
*Cesarean section*	209 (45.5%)	250 (54.5%)	
*Instrumental* *birth*	29 (48.4%)	34 (54%)	
Postpartum grief/anxiety			
*Yes*	315 (46.5%)	363 (53.5%)	0.226
*No*	776 (49.2%)	800 (50.8%)	

**Table 3 healthcare-11-01568-t003:** Quality of life in Group 1 and Group 2.

		Group 1	Group 2	*p*-Value
		*n* = 1091	*n* = 1163	
QOL Child Care	Median (IQR) ^1^	27 (24–29)	25 (23–28)	<0.001
(PQOL scoring)	Mean (±SD)	26.63 ± 3.69	25.63 ± 3.64	
QOL Physical Well-Being	Median (IQR)	33 (30–35)	30 (28–33)	<0.001
(PQOL scoring)	Mean (±SD)	32.60 ± 4.08	29.81 ± 5.93	
QOL Psychological	Median (IQR)	30 (28–31)	27 (25–31)	<0.001
Well-Being (PQOL scoring)	Mean (±SD)	29.52 ± 1.73	26.93 ± 6.12	
QOL Social Inclusion and Support (PQOL scoring)	Median (IQR)	42 (38–46)	36 (33–40)	<0.001
	Mean (±SD)	41.62 ± 5.64	35.36 ± 8.30	

^1^ The values represent mean ± standard deviation or median (IQR) values. IQR, interquartile range; QOL, quality of life; PQOL, perceived quality of life scale.

**Table 4 healthcare-11-01568-t004:** Quality of life and duration of antenatal classes (linear regression).

	Unstandardized Coefficient B	Standardized Coefficient Beta	95% CI for B	*p*-Value
**QOL–Child Care**				
Duration of antenatal classes (min)	0.028	0.333	0.024–0.030	<0.001
**QOL—Physical Well-Being**				
Duration of antenatal classes (min)	0.038	0.463	0.035–0.041	<0.001
**QOL—Psychological Well-Being**				
Duration of antenatal classes (min)	0.037	0.476	0.034–0.039	<0.001
**QOL—Social Inclusion and Support**				
Duration of antenatal classes (min)	0.051	0.606	0.047–0.055	<0.001

CI, confidence interval; QOL, quality of life.

**Table 5 healthcare-11-01568-t005:** Multinominal linear regression: Child Care quality of life and other parameters.

	QOL Child Care	B	Exp(B)	95% CI for Exp(B)	*p*-Value
1	Duration of antenatal classes	−0.004	0.996	0.993–0.998	<0.001
	Age	−0.029	0.972	0.931–1.014	0.186
	Level of education	−0.110	0.896	0.728–1.102	0.299
	Marital status	0.102	1.107	0.911–1.346	0.305
	Economic status	0.062	1.064	0.879–1.288	0.525
	Monthly household income	0.031	1.032	0.824–1.293	0.785
	Place of residence	−0.390	0.677	0.463–0.988	0.043
	Consecutive birth	0.173	1.188	0.913–1.548	0.199
	Type of birth	−0.081	0.922	0.615–1.384	0.696
	Father at birth	−0.089	0.914	0.554–1.509	0.726
3	Duration of antenatal classes	0.002	1.002	1.001–1.003	0.005
	Age	−0.009	0.991	0.975–1.008	0.322
	Level of education	−0.117	0.890	0.803–0.986	0.026
	Marital status	−0.086	0.917	0.834–1.009	0.076
	Economic status	−0.068	0.935	0.846–1.033	0.186
	Monthly household income	−0.033	0.968	0.865–1.083	0.568
	Place of residence	0.204	1.227	1.017–1.479	0.033
	Consecutive birth	−0.097	0.907	0.790–1.042	0.17
	Type of birth	−0.222	0.801	0.653–0.983	0.053
	Father at birth	−0.063	0.939	0.724–1.218	0.635
4	Duration of antenatal classes	0.005	1.005	1.001–1.009	0.018
	Age	−0.001	0.999	0.972–1.028	0.958
	Level of education	−0.033	0.719	0.517–1.000	0.05
	Marital status	−0.003	0.997	0.732–1.359	0.985
	Economic status	−0.192	0.825	0.587–1.160	0.269
	Monthly household income	0.091	1.096	0.768–1.562	0.614
	Place of residence	0.131	1.34	0.630–2.064	0.045
	Consecutive birth	−0.490	0.612	0.358–1.048	0.074
	Type of birth	0.179	1.196	0.696–2.055	0.517
	Father at birth	−1.568	0.209	0.049–0.884	0.053

CI, confidence interval; QOL, quality of life; Ref. Cat. No. 2.

**Table 6 healthcare-11-01568-t006:** Multinominal linear regression: Physical Well-Being quality of life and other parameters.

	QOL Physical	B	Exp(B)	95% CI for Exp(B)	*p*-Value
1	Duration of antenatal classes	−0.026	0.974	0.965–0.983	<0.001
	Age	−0.043	0.958	0.910–1.009	0.103
	Level of education	−0.201	0.819	0.618–1.085	0.164
	Marital status	0.073	1.075	0.840–1.377	0.565
	Economic status	−0.438	0.550	0.247–0.928	<0.001
	Monthly household income	0.232	1.261	0.944–1.685	0.116
	Place of residence	0.012	1.012	0.623–1.643	0.962
	Consecutive birth	0.397	1.487	1.088–2.033	0.013
	Type of birth	−0.535	0.586	0.305–1.124	0.108
	Father at birth	−0.414	0.661	0.328–1.332	0.247
2	Duration of antenatal classes	−0.004	0.996	0.995–0.997	<0.001
	Age	0.002	1.002	0.994–1.010	0.677
	Level of education	0.001	1.001	0.907–1.105	0.983
	Marital status	0.027	1.028	0.937–1.127	0.561
	Economic status	−0.123	0.885	0.801–0.977	0.016
	Monthly household income	0.154	1.167	1.048–1.300	0.055
	Place of residence	−0.054	0.947	0.791–1.134	0.554
	Consecutive birth	0.171	1.187	1.042–1.351	0.01
	Type of birth	−0.015	0.985	0.814–1.193	0.878
	Father at birth	0.043	1.044	0.816–1.335	0.733
4	Duration of antenatal classes	0.004	1.004	1.000–1.008	0.05
	Age	0.008	1.008	0.995–1.021	0.242
	Level of education	−0.21	0.811	0.600–1.095	0.172
	Marital status	−0.182	0.834	0.630–1.103	0.203
	Economic status	0.223	1.249	0.960–1.622	0.095
	Monthly household income	−0.044	0.957	0.683–1.341	0.798
	Place of residence	0.668	1.95	1.091–3.484	0.024
	Consecutive birth	−0.022	0.978	0.652–1.469	0.916
	Type of birth	0.221	1.247	0.735–2.118	0.413
	Father at birth	0.191	1.21	0.589–2.486	0.604

CI, confidence interval; QOL, quality of life; Ref. Cat. No. 3.

**Table 7 healthcare-11-01568-t007:** Multinominal linear regression: Psychological Well-Being quality of life and other parameters.

	QOL Psychological Well Being	B	Exp(B)	95% CI for Exp(B)	*p*-Value
1	Duration of antenatal classes	−0.002	0.998	0.997–0.999	0.003
	Age	−0.044	0.957	0.909–1.008	0.096
	Level of education	−0.197	0.821	0.636–1.059	0.129
	Marital status	−0.021	0.979	0.774–1.238	0.859
	Economic status	−0.596	0.515	0.460–1.258	<0.001
	Monthly household income	0.131	1.140	0.868–1.498	0.346
	Place of residence	−0.300	0.741	0.469–1.170	0.198
	Consecutive birth	0.423	1.527	1.125–2.073	0.007
	Type of birth	−0.416	0.66	0.366–1.188	0.166
	Father at birth	−0.832	0.435	0.212–0.894	0.024
3	Duration of antenatal classes	0.001	1.001	1.001–1.002	<0.001
	Age	−0.003	0.997	0.990–1.005	0.515
	Level of education	−0.055	0.947	0.852–1.052	0.307
	Marital status	−0.058	0.944	0.855–1.042	0.255
	Economic status	0.208	1.231	1.102–1.375	<0.001
	Monthly household income	−0.065	0.937	0.836–1.051	0.27
	Place of residence	−0.055	0.946	0.780–1.147	0.573
	Consecutive birth	−0.043	0.958	0.833–1.102	0.549
	Type of birth	−0.007	0.993	0.811–1.216	0.949
	Father at birth	−0.418	0.659	0.513–0.846	0.053

CI, confidence interval; QOL, quality of life; Ref. Cat. No. 2.

**Table 8 healthcare-11-01568-t008:** Multinominal linear regression: Social Inclusion and Support quality of life and other parameters.

	QOL Social Inclusion and Support	B	Exp(B)	95% CI for Exp(B)	*p*-Value
1	Duration of antenatal classes	−0.027	0.974	0.961–0.986	<0.001
	Age	−0.066	0.936	0.892–0.983	0.058
	Level of education	−0.052	0.949	0.731–1.232	0.694
	Marital status	0.078	1.081	0.857–1.363	0.512
	Economic status	−0.452	0.572	0.282–0.927	<0.001
	Monthly household income	0.149	1.161	0.886–1.520	0.278
	Place of residence	−0.063	0.939	0.598–1.474	0.784
	Consecutive birth	0.378	1.459	1.088–1.957	0.052
	Type of birth	−0.293	0.746	0.418–1.334	0.323
	Father at birth	−0.746	0.474	0.239–0.941	0.033
2	Duration of antenatal classes	0.001	1.001	0.998–1.004	0.575
	Age	−0.009	0.991	0.946–1.038	0.700
	Level of education	−0.070	0.933	0.737–1.180	0.562
	Marital status	0.171	1.187	0.943–1.494	0.144
	Economic status	−0.023	0.978	0.780–1.225	0.844
	Monthly household income	−0.342	0.711	0.543–0.930	0.013
	Place of residence	−0.153	0.858	0.560–1.315	0.482
	Consecutive birth	0.135	1.145	0.840–1.560	0.391
	Type of birth	0.231	1.259	0.823–1.926	0.289
	Father at birth	0.453	1.573	0.967–2.557	0.068
4	Duration of antenatal classes	0.009	1.009	1.008–1.011	<0.001
	Age	0.008	1.008	0.998–1.018	0.106
	Level of education	−0.056	0.946	0.852–1.050	0.294
	Marital status	−0.174	0.840	0.761–0.927	0.051
	Economic status	−0.016	0.985	0.888–1.092	0.768
	Monthly household income	0.158	1.171	1.043–1.314	0.007
	Place of residence	0.130	1.139	0.940–1.381	0.185
	Consecutive birth	−0.147	0.863	0.749–0.996	0.044
	Type of birth	0.111	1.117	0.912–1.369	0.283
	Father at birth	−0.409	0.664	0.503–0.876	0.054
5	Duration of antenatal classes	0.015	1.015	1.010–1.019	<0.001
	Age	−0.014	0.987	0.924–1.053	0.683
	Level of education	−0.231	0.794	0.599–1.052	0.108
	Marital status	−0.102	0.903	0.693–1.177	0.45
	Economic status	−0.052	0.950	0.718–1.256	0.718
	Monthly household income	0.284	1.328	0.974–1.811	0.073
	Place of residence	0.237	1.268	0.756–2.125	0.368
	Consecutive birth	−0.047	0.954	0.640–1.422	0.817
	Type of birth	0.006	1.006	0.579–1.747	0.983
	Father at birth	−0.441	0.643	0.278–1.487	0.302

CI, confidence interval; QOL, quality of life; Ref. Cat. No. 3.

**Table 9 healthcare-11-01568-t009:** Child Care, Physical Well-Being, Psychological Well-Being, Social Inclusion and Support quality of life correlated with the duration of antenatal classes.

Correlation	Spearman Rho	*p*-Value
QOL Child Care—Duration of antenatal classes (min)	0.421	<0.001
QOL Physical Well-Being—Duration of antenatal classes (min)	0.242	<0.001
QOL Psychological Well-Being—Duration of antenatal classes (min)	0.206	<0.001
QOL Social Inclusion and Support—Duration of antenatal classes (min)	0.433	<0.001

QOL, quality of life.

## Data Availability

Data will be available by the authors.

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
