# Peer review of "Association of Maternal Antenatal Education with Quality of Life after Childbirth in the Slovenian Population before and during the COVID-19 Pandemic"

_healthcare, 2023, doi:10.3390/healthcare11111568_

Round 1
Reviewer 1 Report
Thank you for the opportunity to review this paper which has shown the benefit of antenatal education. The paper aimed to assess difference in quality of life after childbirth before and during the pandemic in Slovenia. Findings point to an increased duration of antenatal education associated with a better quality of life after childbirth.
The paper uses appropriate methodology but requires some rewriting to make it flow better. I have a few comments which I think will help. Some of these may relate to translation issues where the word may need changing to make it clearer to the reader.
Title
Line 1. I am unsure what is meant by ‘material antenatal education’. Is there a better way to describe this to assist international readers.
Abstract
The abstract reflects the contents of the paper.
1) Line 17. I am not sure I understand this sentence, ‘Based on an online survey, 2 groups were interviewed. When I read the rest of the paper it looks like the questionnaire and interview are the same. Was the questionnaire prior to birth and the interview afterwards? Please could you clarify this here and in the methods section.
2) Line 18. The word ‘birth’ is more commonly used than ‘delivery’ in papers. I would suggest changing this word throughout.
Background
3) Line 34. I would suggest changing ‘himself’ to ‘themselves’.
4) Lines 34-35. Using the words unprepared and untrained seems a very strong statement to make without a reference. Not all couples go to antenatal classes and I would not expect this to result in ‘numerous personality and partnership problems…….’. May be a sentence which states that ‘sometimes couples without antenatal education/preparation can lead to….’. This will need to be referenced if this is a research finding.
5) Line 80. I am not sure what is meant by ‘field of psychosocial perception..’. Could this be reworded for clarity.
6) Line 99-100. I am not sure what ‘the described area…’ relates to.
Methods
I found the methods section did not flow well. Could you restructure the section up to the paragraph about ethics. Possibly adding subheadings may make it clearer. The materials and methods and recruitment seem to be mixed up. I have made some comments about the areas that I am struggling to understand.
7) Line 105. I do not think the words ‘in the research work…’ are necessary. Also is the word ‘working’ needed. Could the line be replace with, ‘This study used mixed methods design…’
8) Line 107. Here you mention the PQOL questionnaire and again in 132. Why is it in twice.
9) Line 115. You mention the interview with organisers of antenatal classes and again in 144 you say the same. Are these two separate groups of organisers or the same. If they are the same, does this need to be repeated, could you remove one of the sentences.
10) Line 121. I am not sure what you mean by mothers being interviewed with an online questionnaire. Interviews and questionnaires are different. Could you make it clear if it was a self completed questionnaire or an interview. If it was an interview who conducted the interview.
11) Line 121. Who were the mothers that were interviewed. Are these the same as the 15 in the pilot study (line122) or the 2254 in line 126?
The paragraphs are not flowing logically.
12) Line 121. You describe a questionnaire and its reliability. Then line 132 you talk about how many women participate. Then 132 you mention a questionnaire again. I am not sure if this is the same questionnaire in line 122 and 131?
13) Line 157. I am not sure why this sentence is in a single paragraph, could it be added to the one before.
There also needs to be more information about the recruitment process. This is not replicable.
14) How were women identified, how did they find out about the online questionnaires, how many were invited and how many responded.
15) Organisers: Who, how were they identified, how many were invited to take part, how many took part? What were they asked, how long did the interviews last? Who interviewed the organisers?
Results
16) I may have missed the information but is there data which shows the duration of antenatal classes each person took or how many they attended. Did all women attend all classes or was there a range of attendance, was this taken into account?
17) Line 224. Improvement over the null? There is no mention of a null hypothesis in this study. What are you referring to here.
Discussion
Overall I find the discussion has a lot of previous research but without clearly relating this to your own and discussing why it is different or similar. Could you relook at this section. I am having to reread it several times to try and understand what you are saying. The paragraphs are quite long; shorter paragraphs with a single finding would be easier to understand.
18) Lines 135. Antgenatal stress? Do you mean antenatal.
19) Lines 309-330. This paragraph is very long and I am struggling to understand why all you have added another list of previous research to the discussion section. Could a few of these be removed unless they are being used to illustrate your research. This sections sounds more like a background paragraph.
20) The second paragraph again is very long. It starts by discussing duration of antenatal classes, then levels of education, income and support, then the benefits of AN classes and ends discussing self-efficacy. Could this be broken up into smaller paragraphs and how each of these relate to the duration of classes.
21) Lines 351 and 372 are both discussing the benefits of AN classes and who information is provided and how this helps mothers. Is there a distinction between these two or is it repetition?
22) Lines 308 and 392 are saying the same. Do you need both?
Conclusion
The conclusion would benefit from rewriting to make it clearer and correct the grammar. I am not sure why you are talking about previous research, this paragraph should conclude your findings and what this study adds to previous knowledge.
23) Line 402. I do not think the word assume is the correct word to use here. You could leave out the words ‘We assume that..’ and start the sentence ‘Our research….’
References
Appropriate.
Overall the quality of English is good.
There are a few areas where i have suggested rewriting sections to make the sentences and a few words that could be replaced or i have not understood what is being said.
Author Response
We appreciate the reviewer's comments and suggestions. We believe that we have managed to fully address the reviewers’ comment and hope that this version of the manuscript is improved. Point by point response to comments is in attached file.

Reviewer 2 Report
This paper presents interesting and valuable results.
There are some suggestions for improvement:
There need to be citations in the first paragraph for the information on personality and partnership problems and the information about expectations.
2) In the second paragraph, the frequency of antenatal classes should be given.
3) It is unclear in the third paragraph how community service affects quality of life and why this information is relevant to the paragraph and paper as a whole.
4) In the Materials and Methods section, it is unclear what the topics of the online questionnaire were other than the PQOL. These need to be explained with example items.
5) Example items for all subscales of the PQOL should be given.
6) The points on the Likert scale for the PQOL should be reported.
7) In Table 3, means in addition to medians should be reported.
8) In the linear regression model, it states that for each additional minute of antenatal classes, quality of life increased by .28, .37, etc. when the B weights were .028 and .037 – therefore, the quality of life increased by .028 and .037.
9) Given the large sample size, it is expected that B weights of small magnitude will be statistically significant. The small magnitude of these results should be discussed as a limitation.
10) Dividing the PQOL subscales into 5 categories did not seem to work well, as there was an insufficient sample size for several categories. Another number of groupings could potentially work better.
11) Dividing PQOL total scores into categories is not a good strategy for data analysis, as it discards variance when breaking down totals.
12) Different reference categories were used for some of the different PQOL subscale analyses, which makes them not comparable.
13) The predictor in the multinomial logistic regression showing a relationship between duration and quality of life had a magnitude of .002 and was likely statistically significant due to the sample size. This fact should be mentioned as a limitation.
14) In the first paragraph of the Discussion, in-person vs. online classes were mentioned. Were pre-COVID antenatal classes in person and the ones during COVID held online? If so, this difference and how it affects the results should be discussed.
15) In the second paragraph of the Discussion, research supporting the relationships with demographic variables should be included.
16) The second paragraph covers demographics, usefulness of antenatal classes, and psychological correlates of pregnancy. These are all different topics that should be in different paragraphs.
17) Line 374 states that the antenatal classes made mothers feel more prepared and confident. The results are only about duration and do not address these variables, so this claim cannot be made.
18) Random sampling is listed as a limitation when it is preferred to non-probability sampling methods.
19) The limitation of not being able to generalize results beyond Slovenia to other cultures and countries should be mentioned.
There were a few places where the word "the" was missing before a noun.
Author Response

(The authors gave the same response as above.)

Reviewer 3 Report
Dear Authors,
you have presented interesting and important topic in your manuscript, but I have some comments and suggestions about it.
Please check the Microsoft Word template to prepare your manuscript as it is required by the Journal.
The abstract should be without headings.
Please chcek carefully whole your manuscript about references in text for exaple line 44 or at the end of first paragraph on page 2 there should be references.
Section Materials and Methods - it would be more clear for Readers if it was divded for subsections for exaple: Study Design and Participants, Statistical Analysis. Also try to sort all the you presented information in Materials an Methods but by when and where, who (inclusion and exclusion criteria, how many was involved in the study and how many was rejected), how did you conduct the study and what kind of tools you used for easier follow of study by Readers.
Section Results - please modify table 1 it is hard to interpretate it - put the legend into table - if you need help check articles published in Healthcare (for example https://www.mdpi.com/2227-9032/11/9/1357)
Table 2 presents information about sample size - maybe it should be in the section Materials and methods.
Description of tables 5 to 8 are not clear - what are 5 categories, what about predictors what are the references for them?
Author Response

(The authors gave the same response as above.)

Round 2
Reviewer 1 Report
Thank you for taking the suggestion and making changes. The manuscript is now much easier for the reader. There is only one final comment about my previous review:
My comment: Line 18. The word ‘birth’ is more commonly used than ‘delivery’ in papers. I would suggest changing this word throughout.
Your reply: The word “birth” has been changed to “delivery” throughout the text
My comment may not have been clear. The word 'birth' is more commonly used. I would suggest the word 'delivery' is changed to 'birth', not birth to delivery. Sorry for any confusion. It may be manuscript specific, I have asked the editor for clarity to see if they have a specific suggestion.
Reviewer 2 Report
Most reviewer comments were answered adequately. However, no example items were given for the PQOL. Information was added about virtual classes, but it was still not clearly stated that the 2018 classes were in-person and that the differences might affect the results.
